# Anchor-Guided Behavior Cloning with Offline Reinforcement Learning for Robust Autonomous Driving

## Abstract

Learning robust driving policy from logged data is challenging due to the distribution shift between open-loop training and closed-loop deployment. We propose ABC-RL, a hybrid framework that integrates Anchor-guided Behavior Cloning (ABC) with offline Reinforcement Learning (RL) under a single-step world model to address this issue. A key innovation of our method is anchor-based behavior cloning, which introduces dynamics-aware intermediate trajectory targets. These anchor points normalize trajectories across different speeds and driving styles, enabling more accurate trajectory prediction and improving generalization to diverse driving scenarios. In addition, we leverage a learned world model to support offline RL: given the current state and action, the world model predicts the next state, which is then encoded to estimate the reward, allowing effective policy learning without environmental interaction. This model-assisted training process enhances learning efficiency and stability under offline settings. To evaluate the effectiveness of ABC-RL, we perform open-loop assessments and develop a closed-loop simulation benchmark using the nuScenes dataset, enabling a comprehensive evaluation of planning stability and safety. Our method achieves state-of-the-art performance, significantly outperforming behavior cloning baselines in both open-loop and closed-loop evaluations. Notably, ABC-RL reduces open-loop trajectory error from 0.29 m to 0.22 m and reduces closed-loop collision rates by over 57%, demonstrating the practical benefits of integrating trajectory-level supervision with model-assisted offline policy refinement. Our findings highlight the potential of ABC-RL under learned world models, offering a scalable and robust solution for real-world autonomous driving.

## 1 Introduction

Learning a scalable end-to-end driving policy from offline datasets significantly appeals to autonomous vehicles. Imitation Learning (IL) (Codevilla et al., 2018; Zhang et al., 2021; Vitelli et al., 2022; Bojarski et al., 2016) is a popular strategy that leverages human driving logs to train policies. Behavior Cloning (BC), a specific approach within this paradigm, is widely adopted for its simplicity in implementation. However, this method often suffers from compounding errors when deployed, as minor deviations from the expert trajectory can accumulate and lead the policy into out-of-distribution states. A well-known challenge in BC is the discrepancy between open-loop training (evaluation on logged data) and closed-loop deployment (autonomous rollout) (Duc et al., 2021; Ross et al., 2011; Ng & Russell, 2000). Pure BC can achieve high open-loop accuracy, especially in common-case scenarios. Still, it lacks robustness in closed-loop driving, where the vehicle must recover from its mistakes (Zhang et al., 2024). On the other hand, Reinforcement Learning (RL) can improve robustness by optimizing long-term rewards. Still, online RL requires interactive environments or simulators and can produce unnatural behaviors if not carefully constrained (Kiran et al., 2022). Recent work (Gao et al., 2025) has shown that building photorealistic simulators (e.g., with 3D Gaussian Splatting (Kerbl et al., 2023) ) enables closed-loop RL training for driving, often combining a BC objective to maintain human-like behavior. However, such approaches require complex infrastructure and still face challenges with stability and efficiency (Ljungbergh et al., 2024).

This motivates the need for a practical framework that gains the benefits of RL without expensive online interaction while preserving the safe driving behavior of BC. This paper proposes a novel anchor-guided BC with an offline RL framework to address these challenges. We integrate BC and offline RL into a unified framework, using a single-step world model to simulate environment transitions from logged data (Wang et al., 2025). The core innovation is a dynamics-aware anchor point representing the vehicle's future trajectory. Instead of directly predicting raw future positions, our policy predicts anchor points normalized by the ego dynamics, serving as a temporally adaptive goal (Li et al., 2021). By conditioning on speed, the network learns a trajectory representation invariant to velocity scaling – effectively normalizing trajectory prediction across different driving speeds. This makes learning easier and yields significantly better accuracy in matching expert trajectories. Moreover, the learned world model allows us to incorporate an offline reinforcement learning (RL) objective. By combining the learned transition dynamics with a reward prediction head, we can refine the policy beyond mere behavioral cloning. The RL component leverages the fixed offline dataset to gently guide the policy toward higher cumulative returns—such as smoother and safer driving maneuvers—without needing online environment interaction. Importantly, our approach maintains the reliability of imitating familiar behaviors while enhancing robustness in novel or unfamiliar scenarios through value-based feedback.

We evaluate our approach on the nuScenes dataset (Caesar et al., 2020), which provides real-world driving logs with complex traffic scenarios. Empirical results show that our hybrid method achieves state-of-the-art open-loop performance, outperforming strong BC baselines in accuracy by a large margin. More importantly, when evaluating in closed-loop simulations, our learned policy exhibits substantially enhanced stability and safety: collision rates are notably reduced compared to policies trained with BC alone. For example, our method reduces the average L2 trajectory error from 0.29 m to 0.22 m (mean Euclidean distance between the predicted and ground-truth future trajectory). At the same time, it lowers closed-loop collision events by over 57% compared to a pure BC baseline, demonstrating significantly improved safety during deployment. These gains underscore the practical impact of combining BC and offline RL – our policy not only mimics the expert more faithfully but also handles out-of-distribution events more properly. Our work is the first to successfully unify BC and offline RL in an end-to-end driving context with a learned world model, demonstrating a viable path toward accurate and robust autonomous driving without resorting to online data collection. We hope our approach contributes to bridging the gap between research and reliable real-world deployment of learned driving policies.

Our main contributions are summarized as follows:

1. We propose a unified ABC-RL framework that combines BC with offline RL under a single-step world model. This framework enables reward-guided policy optimization using logged data and supports closed-loop training on the nuScenes dataset.

2. We introduce an anchor-based BC method that leverages constant-velocity anchor points as intermediate targets for predicting 3-second future trajectories, improving temporal consistency and prediction accuracy.

3. Our method achieves strong experimental metrics, reducing open-loop L2 prediction error from 0.29 m to 0.22 m and significantly reducing closed-loop collision rates by over 57%.

## 2 RELATED WORK

**Behavior Cloning for Autonomous Driving**

Researchers have widely adopted Behavior Cloning (BC) for autonomous driving, where end-to-end models learn to map sensor inputs to future trajectories or control commands by mimicking expert demonstrations. Recent advances in BC incorporate increasingly structured scene understanding. For example, UniAD (Hu et al., 2023) proposes a unified framework that integrates perception, prediction, and planning, while TCP (Wu et al., 2022) scores candidate trajectories to improve decision-making. VAD (Jiang et al., 2023) further introduces a vectorized bird's-eye view (BEV) representation that encodes rich scene geometry, enhancing the model's ability to capture complex traffic scenarios. Surprisingly, a later study AD-MLP (Zhai et al., 2023) demonstrated that even a simple multi-layer perceptron (MLP) model using only ego-vehicle motion (e.g., velocity, acceleration) can achieve strong performance on the nuScenes open-loop benchmark. Another recent study (Li et al., 2024)

highlights the need for more diverse and challenging evaluation protocols to reflect real-world complexity and generalization requirements better.

Despite their success in open-loop evaluations, BC methods often suffer from distribution shift during closed-loop deployment, where small prediction errors can accumulate and lead the agent into unfamiliar states. These out-of-distribution scenarios are rarely encountered in training data, making pure BC policies fragile when required to recover from their own mistakes. Additionally, BC lacks a mechanism to reason about long-term consequences or trade-offs, limiting its effectiveness in complex or ambiguous driving situations.

Our work builds on the VAD framework's scene encoding capabilities and addresses the above limitations by integrating offline reinforcement learning. This hybrid approach enhances closed-loop robustness by leveraging a learned world model to simulate future outcomes and refine the policy under distribution shift.

**Reinforcement Learning in Driving**

Reinforcement Learning (RL) optimizes long-term rewards and can handle rare events beyond the training distribution. RAD (Gao et al., 2025) employs online RL with a 3D Gaussian Splatting-based simulator (Kerbl et al., 2023) to train recovery behaviors, combining BC for human-likeness (Hester et al., 2018; Rajeswaran et al., 2018; Vecerik et al., 2018). Other methods explore real-vehicle safe RL (Wen et al., 2020) and crash severity prediction for reward shaping (Holen et al., 2024). Despite effectiveness, online RL demands heavy computation and complex reward design (Knox et al., 2024; Pan et al., 2017; Liang et al., 2018; Volodymyr et al., 2015).

Offline RL improves policies from fixed logs without requiring simulators (Kostrikov et al., 2022; Kumar et al., 2020), but suffers from distributional shift due to limited state-action coverage (An et al., 2021; Yeom et al., 2024; Fujimoto et al., 2019). Recent efforts such as (Lee & Kwon, 2025) combine offline RL with episodic future thinking to enhance adaptivity.

Our method belongs to this family but introduces a key distinction: we leverage a learned world model trained on nuScenes to simulate state transitions. This allows the policy to explore novel actions and observe their outcomes beyond what is directly available in the offline data.

**World Models and Latent Policy Learning**

World models aim to simulate environment dynamics in a compact latent space, enabling agents to learn policies without interacting with the real environment or simulator during training. This idea underpins model-based RL frameworks like DreamerV3 (Hafner et al., 2025), which learn latent dynamics and train policies entirely within a learned world, achieving strong performance in continuous control tasks.

This paradigm has been extended in autonomous driving to address partial observability and semantic complexity. Doe-1 (Zheng et al., 2024) constructs a recurrent latent dynamics model conditioned on visual inputs to support closed-loop planning from camera observations. OccLLaMA (Wei et al., 2024) develops a generative occupancy-based world model enriched by language and multimodal priors. Yang et al. (Yang et al., 2024) propose a vision-centric world model that forecasts occupancy grids for downstream planning.

These approaches demonstrate the feasibility of long-horizon policy learning under uncertainty through expressive latent dynamics. However, they often require significant model complexity and training costs.

## 3 METHOD

We propose Anchor-guided Behavior Cloning with offline Reinforcement Learning (ABC-RL). This hybrid framework (Fujimoto & Gu, 2021) combines anchor-guided BC and offline RL (Fujimoto et al., 2018) within a single-step world model to train robust driving policies from offline data. In broad strokes, this integrated approach uses anchor-guided BC to initialize the policy, leverages a learned world model to simulate diverse environment dynamics for enhanced generalization, and applies offline RL (TD3 (Fujimoto et al., 2018) +BC) to refine the policy under static dataset constraints. Together, these components form a cohesive training pipeline that capitalizes on the complementary

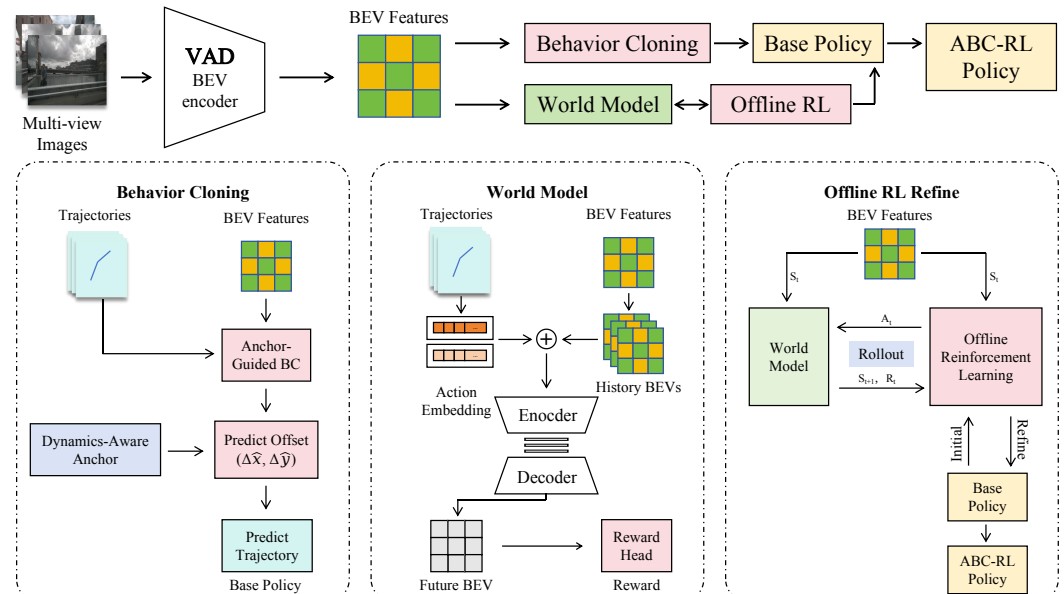

Figure 1: **Overall architecture of the ABC-RL framework.** The ABC-RL framework combines an anchor-guided BC module initialized via BC, a single-step world model simulating environment transitions with BEV features and Transformer-based spatiotemporal fusion, and an offline RL pipeline using TD3+BC with conservative constraints to refine policies through synthetic world model rollouts while maintaining behavioral fidelity.

strengths of BC and reinforcement learning, and the following subsections detail each element of our method in Figure 1.

## 3.1 ANCHOR-GUIDED BEHAVIOR CLONING

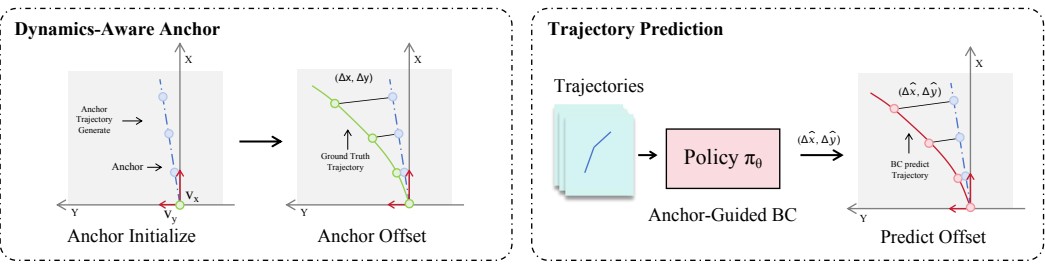

Figure 2: **Anchor-guided behavior cloning module.** The policy predicts velocity-normalized trajectory offsets relative to constant-velocity anchors initialized by BC. The BC module reconstructs future trajectories as $\hat{x}_k = x_k^{anc} + \Delta\hat{x}_k$ and $\hat{y}_k = y_k^{anc} + \Delta\hat{y}_k$

We first train a policy via BC using anchor-guided trajectory prediction, as shown in Figure 2. The anchor trajectory is constructed by assuming the ego vehicle continues with its current velocity $(v_x, v_y)$ in the ego-centric coordinate frame. For a planning horizon of $H$ steps with time interval $\Delta t$, the anchor trajectory $(x_k^{anc}, y_k^{anc})$ is defined as:

$$x_k^{anc} = x_0 + v_x k\Delta t, \qquad y_k^{anc} = y_0 + v_y k\Delta t, \tag{1}$$

for $k = 1, 2, \ldots, H$, where $(x_0, y_0)$ is the current position and $(v_x, v_y)$ is the current velocity of the ego vehicle in the ego-centric frame.

We define the trajectory offset between the ground-truth future trajectory $(x_k^{gt}, y_k^{gt})$ and the anchor as:

$$\Delta x_k = x_k^{gt} - x_k^{anc}, \qquad \Delta y_k = y_k^{gt} - y_k^{anc}. \tag{2}$$

The policy $\pi_\theta$ is trained to predict these offsets based on the current state, simplifying the learning problem by normalizing for ego-motion. For each training sample, the predicted future trajectory is reconstructed as follows:

$$\hat{x}_k = x_k^{anc} + \Delta\hat{x}_k, \qquad \hat{y}_k = y_k^{anc} + \Delta\hat{y}_k, \quad \text{for} \quad k = 1, \ldots, H. \tag{3}$$

The BC loss minimizes the mean squared error between the predicted and ground-truth trajectories:

$$\mathcal{L}_{BC}(\theta) = \frac{1}{H} \sum_{k=1}^{H} \left[ (\hat{x}_k - x_k^{gt})^2 + (\hat{y}_k - y_k^{gt})^2 \right]. \tag{4}$$

This anchor-guided formulation provides a motion-normalized prediction target, making trajectory learning more stable and generalizable across varying speeds.

## 3.2 WORLD MODEL

We also train a world model to simulate the environment's dynamics. The world model (typically a neural network) learns to predict state transitions and rewards from the offline data, defined as $f_\phi : (S_t, a_t) \mapsto (s_{t+1}, r_t)$, where $S_t$ is the set of historical BEV features, $a_t$ represents the ego's actions at time $t$, $s_{t+1}$ is the BEV features at time $t + 1$, and $r_t$ is the reward obtained when the RL agent takes action $a_t$ at time $t$. The world model operates through three key stages: BEV feature encoding, Spatiotemporal fusion via Transformer (Vaswani et al., 2017), and future state decoding.

**Inputs and Preprocessing of World Models:** Given $N$ historical trajectories $A_{prev} = \{a_t, a_{t-1}, a_{t-2}, \ldots, a_{t-N}\}$ and historical BEV features $s_{prev} = \{s_t, s_{t-1}, s_{t-2}, \ldots, s_{t-N}\}$, the historical BEV features are mapped to the current BEV feature $s_t$ via coordinate transformation. Then the input $S_t \in \mathbb{R}^{hisframes \times h \times w \times c}$ of the world model is obtained. Subsequently, historical feature fusion is conducted through the convolutional layer.

**BEV Feature Encoding:** The fused historical BEV features are downsampled by an encoder equipped with a Convolutional Block Attention Module (CBAM) (Woo et al., 2018) to generate a compact latent representation $F_t$, which retains spatial relationships and semantic features.

**ActionSpatial Fusion via Transformer:** The action of the RL agent $a_t$ is encoded into an action embedding $A_t \in \mathbb{R}^c$ via an embedding layer, projected into the latent space through a linear layer and concatenated with flattened BEV features along the spatial dimension. The combined features are denoted as $\hat{F}_t \in \mathbb{R}^{(h/4 \times w/4) \times c}$. Then, a 3-layer Transformer encoder with an 8-head attention mechanism is used to model temporal evolution and spatial interactions. Positional encodings are added to preserve the spatial coordinates of BEV features. The Transformer computes cross-attention between BEV cells and action embeddings, enabling the model to explain how specific actions affect different spatial regions.

**Future State Decoding:** The Transformer output is reshaped back to $h/4 \times w/4$ resolution and processed through two convolution blocks with CBAM attention. Each block upsamples the spatial resolution by $4\times$ while reducing channels, predicting future BEV states $s_{t+1}$.

**Reward Head:** To estimate rewards from predicted BEV states $s_{t+1}$, we employ a lightweight reward head comprising three convolutional blocks and a spatial attention mechanism. The attended features are globally pooled and passed through an MLP to produce 2D reward logits indicating positive (goal-directed) or harmful (unsafe) outcomes. This module is trained using supervised reward labels and supports value learning under the world model's dynamics.

**Integration with Planning System:** The world model can generate plausible future trajectories given a sequence of actions by encoding states into a latent space and learning the environment's transition function. We leverage the world model as a surrogate environment for planning and evaluation in the offline setting. In essence, the world model enables policy rollouts and value estimation without additional real environment interactions, providing a safe and efficient way to assess and improve policies using only the logged data.

## 3.3 OFFLINE REINFORCEMENT LEARNING

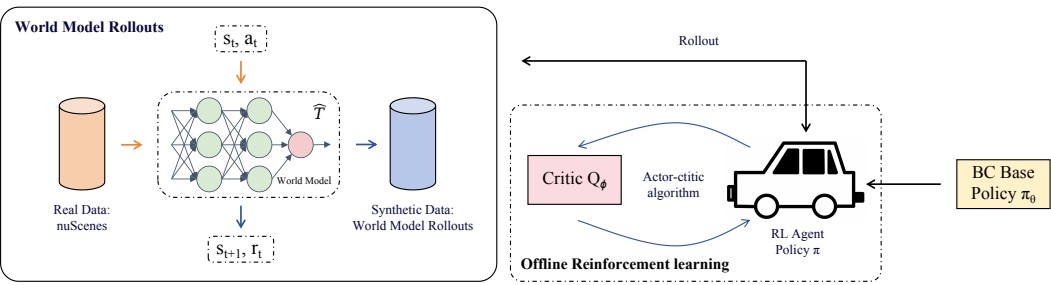

Figure 3: Offline reinforcement learning fine-tuning with world model and BC constraints. Initially initialized by BC, the policy is refined using TD3+BC with conservative constraints. Synthetic trajectories generated by the world model enable offline evaluation of candidate actions while maintaining proximity to the behavioral policy.

As illustrated in Figure 3, we refine the policy network (initialized by the anchor-guided BC model) using an offline reinforcement learning approach. The agent learns from a fixed dataset without additional environment interactions in this stage. To enable temporal-difference updates from static data, we leverage a learned one-step world model $\mathcal{M}$ to simulate the environment dynamics. Instead of a standard TD update, we adopt the Twin Delayed DDPG (TD3) algorithm (Fujimoto et al., 2018), which provides several stability benefits: (i) twin $Q$-networks with a minimum-value target to mitigate overestimation, (ii) target policy smoothing by injecting noise into target actions, and (iii) delayed policy (actor) updates. We integrate these TD3 components into our offline RL framework with a behavior cloning regularizer (TD3+BC (Fujimoto & Gu, 2021)) to constrain the policy toward offline data distribution.

Concretely, we maintain two critic networks $Q_1(s, a)$ and $Q_2(s, a)$ with parameters $\theta_1, \theta_2$, along with corresponding target networks $Q_1', Q_2'$. Each critic takes as input the current BEV state $s$ and a candidate ego-action $a$, then uses the world model $\mathcal{M}$ to predict the next state and reward, i.e., $(\hat{s}', r) = \mathcal{M}(s, a)$. The predicted reward $r$ reflects future outcomes such as goal achievement or potential collisions. The critic networks are trained using the TD3 loss, where the target value is computed with the more conservative of the two target $Q$ networks. Specifically, the target is given by:

$$y = r + \gamma \min_{i=1,2} Q_i'(\hat{s}', \pi'(\hat{s}')), \tag{5}$$

where $\pi'$ is the delayed target policy and $\gamma$ is the discount factor. This formulation allows the Q-networks to learn from imagined transitions provided by the world model, enabling policy improvement without real environment interaction.

$$y(s, a) = r(\hat{s}') + \gamma \min_{i \in \{1,2\}} Q_i'(\hat{s}', \pi'(\hat{s}') + \epsilon), \tag{6}$$

where $\pi'$ is the target policy (actor) network and $\epsilon \sim \mathcal{N}(0, \sigma)$ is a slight clipped noise added to the target action (target policy smoothing). The critic parameters $\theta_i$ are then updated by minimizing the mean-squared Bellman error:

$$\mathbb{E}_{(s,a) \sim \mathcal{D}} \left[ (Q_i(s, a) - y(s, a))^2 \right], \quad \text{for } i \in \{1, 2\}. \tag{7}$$

Using the minimum $Q'$ in Eq. equation 7 (twin-$Q$ min) guards against overestimated value targets, while the injected noise $\epsilon$ smooths the target policy to prevent exploiting narrow peaks in $Q$ estimates.

The policy (actor) network $\pi_\phi$ is updated in a delayed fashion relative to the critics (e.g., one policy update for every two critic updates) to improve training stability. The actor is optimized to maximize the $Q$-value of its actions while remaining close to the behavior policy that generated the offline data. In practice, we employ the TD3+BC strategy, augmenting the deterministic policy gradient objective with a behavior cloning term that penalizes deviation from the dataset actions. Specifically, letting $(s, a)$ be sampled from the offline dataset $D$, the policy update aims to maximize

$\lambda Q_1(s, \pi_\phi(s)) - \|\pi_\phi(s) - a\|_2^2$, where $a$ is the logged expert action for state $s$ and $\lambda$ controls the trade-off between RL and imitation. Equivalently, the actor loss to minimize is given by:

$$\mathcal{L}_{\text{actor}}(\phi) = -\lambda \, \mathbb{E}_{s \sim \mathcal{D}}\big[Q_1(s, \pi_\phi(s))\big] + \mathbb{E}_{(s,a) \sim \mathcal{D}}\big[\|\pi_\phi(s) - a\|_2^2\big], \qquad (8)$$

This loss combines the standard deterministic policy gradient (first term, using $Q_1$ as the critic for gradient computation) with a behavioral cloning error (second term) that keeps the refined policy anchored to the demonstrator behavior. By initializing $\pi\phi$ from the anchor-guided BC model and using a moderate regularization weight $\lambda$, our offline TD3+BC training (Figure 3) effectively improves policy performance (via TD learning of the reward) without straying far outside the support of the offline dataset. The result is a policy refined beyond pure imitation, guided by learned Q-value estimates in the single-step simulated environment while maintaining the safety and generalization benefits of the prior BC initialization.

## 4 EXPERIMENTS

We conduct extensive experiments on the nuScenes dataset to validate the effectiveness of our proposed ABC-RL framework. Our evaluation is into three main parts: (1) open-loop evaluation on logged nuScenes data to assess predictive accuracy and safety, (2) analysis of world model fidelity and its ability to simulate causally consistent future states, and (3) closed-loop evaluation in a realistic simulator built with 3D Gaussian Splatting (3DGS) to test the robustness of the learned policy under interactive deployment. These experiments collectively verify that our method achieves superior planning accuracy and safety compared to behavior cloning baselines and alternative offline reinforcement learning setups.

### 4.1 OPEN-LOOP EVALUATION ON NUSCENES

We evaluate our policy on the nuScenes dataset under the standard open-loop protocol, where the policy is given logged sensor data. We must predict a future trajectory for the ego vehicle without interacting with the environment. We report two primary metrics from the nuScenes planning benchmark: average L2 trajectory error (mean Euclidean distance between the predicted and ground-truth future trajectory) and collision rate (percentage of predicted trajectories that would result in a collision with any obstacles).

As shown in Table 1, our method (Anchor-BC) achieves the best overall performance, reducing the average L2 trajectory error to **0.22 m** while maintaining a competitive collision rate of **0.19%**. Compared to the BC only (AD-MLP, 0.29 m / 0.19%), this reflects substantial improvements in accuracy with similar safety.

The ablated variant (No-Anchor-BC) removes the anchor-guided parameterization and adopts the same direct supervision strategy as AD-MLP. Its comparable performance to AD-MLP indicates that, without anchors, the model only fails to improve over BC. This result highlights the necessity of incorporating constant-velocity anchor points for accurate trajectory prediction and demonstrates their key role in enhancing planning fidelity.

Compared to vectorized models like VAD, which achieve the lowest collision rate (0.14%) but suffer from higher average planning error (0.37 m), our method offers a better trade-off between precision and safety in open-loop planning.

Table 1: Open-loop evaluation on nuScenes.

| Method | L2 Error (m) ↓ | | | | Collision Rate (%) ↓ | | | |
|---|---|---|---|---|---|---|---|---|
| | 1s | 2s | 3s | Avg. | 1s | 2s | 3s | Avg. |
| VAD | **0.17** | 0.34 | 0.60 | 0.37 | **0.07** | **0.10** | 0.24 | **0.14** |
| AD-MLP | 0.20 | 0.26 | 0.41 | 0.29 | 0.17 | 0.18 | 0.24 | 0.19 |
| No-Anchor-BC | 0.22 | 0.29 | 0.39 | 0.30 | 0.18 | 0.22 | 0.28 | 0.23 |
| Anchor-BC | 0.18 | **0.21** | **0.27** | **0.22** | 0.19 | 0.16 | **0.21** | 0.19 |

## 4.2 WORLD MODEL EFFECTIVENESS VALIDATION

We validate the learned world model from two key aspects: *feature-level fidelity* and *causal consistency*, focusing on its ability to produce accurate and action-sensitive BEV features for downstream policy learning.

**(1) Feature-level fidelity.**
We evaluate how closely the predicted BEV features match those generated by a pre-trained VAD-Tiny encoder. A shared decoder extracts object and lane predictions from both sources.

For object detection, we report **mean Average Precision (mAP)** and **nuScenes Detection Score (NDS)**. For lane evaluation, we measure the proportion of predicted lane instances whose average distance to the corresponding ground-truth lane falls below fixed thresholds (e.g., 0.5 m and 1.0 m).

As shown in Table 2, the world model achieves 82.2% and 93.4% of the VAD-Tiny's mAP and NDS scores, respectively. It also maintains over 91% lane alignment accuracy across both thresholds. These results demonstrate that the world model preserves a high degree of perceptual fidelity relative to the original BEV encoder.

Table 2: Perception fidelity: evaluating world model predictions against VAD-Tiny outputs.

| Source | Bbox/mAP | Bbox/NDS | Lane/Th=0.5m | Lane/Th=1.0m |
|---|---|---|---|---|
| VAD-Tiny | 0.2698 | 0.3894 | 0.150 | 0.438 |
| World Model | 0.2217 | 0.3637 | 0.137 | 0.415 |
| Retention | 82.2% | 93.4% | 91.3% | 94.7% |

**(2) Causal response.**
We apply a +3m offset along the x-axis to assess action sensitivity and observe the resulting BEV features. As shown in Figure 4, scene elements shift accordingly, appearing closer to the ego-vehicle, confirming that the world model responds causally to action perturbations.

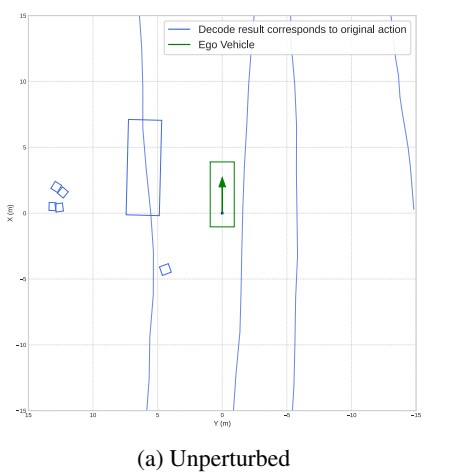
(a) Unperturbed

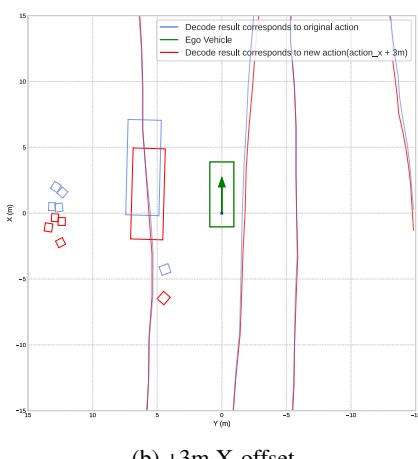
(b) +3m X-offset

Figure 4: Scene shifts forward as expected under a +3m action perturbation.

These results demonstrate that our world model accurately reconstructs scene semantics and captures causal dynamics, supporting safe and generalizable policy learning.

## 4.3 CLOSED-LOOP EVALUATION IN 3DGS SIMULATION

We assess policy performance in a closed-loop setting using a realistic simulator constructed via 3DGS. The simulator reconstructs digital twins of real-world scenes, allowing interaction with other agents in real-time. We report one key metric: collisions per episode (lower is better).

As shown in Table 3, our method (ABC-RL) achieves the lowest collision rate of **0.3%**, significantly outperforming all other approaches, including the human policy (Oracle, 0.7%). This result indicates superior safety and robustness in closed-loop driving.

Interestingly, BC methods—including AD-MLP, VAD, and our Anchor-BC—yield higher or comparable collision rates to human policy. This result suggests a limitation of pure BC: since these models strictly mimic the expert demonstrations, they cannot correct for suboptimal or unsafe behaviors in the dataset.

Notably, VAD performs the worst among all methods. We attribute this to its lack of yaw angle prediction—VAD only predicts the $(x, y)$ coordinates of the future trajectory without modeling orientation changes. As a result, it struggles with sharp turns or directional corrections, leading to more frequent collisions in closed-loop rollouts. In contrast, ABC-RL combines BC with offline RL, enabling the policy to reinforce beneficial behaviors while suppressing risky ones. This method leads to more stable, risk-aware decisions and substantially improved closed-loop performance.

Table 3: Closed-loop evaluation in 3DGS simulator.

| Method | Collisions (%) ↓ |
|---|---|
| Oracle | 0.7 |
| VAD | 0.96 |
| AD-MLP | 0.73 |
| Anchor-BC | 0.75 |
| ABC-RL | **0.3** |

To further illustrate the improvement brought by our method, we present two representative scenarios in Figure 5. In each case, the original action (represented by the red Ego Vehicle) collides with a pedestrian or the curb. By contrast, our policy's adjusted action (shown as the green New Ego Vehicle) successfully avoids risky behavior, demonstrating enhanced safety and rule compliance.

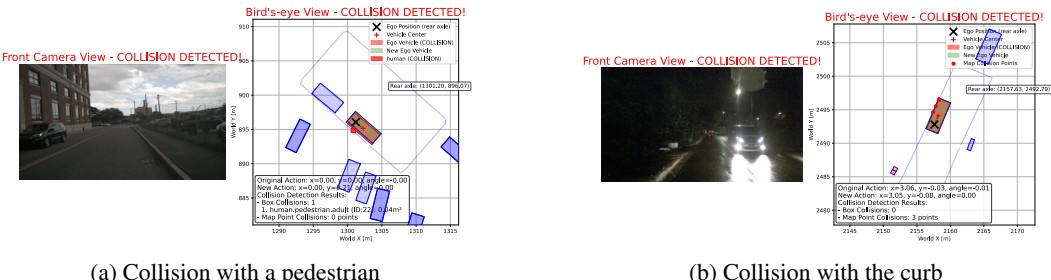

(a) Collision with a pedestrian    (b) Collision with the curb

Figure 5: Visualization results of our method on two representative scenes.

## 5 CONCLUSION

We proposed an anchor-guided offline reinforcement learning framework for autonomous driving that combines behavior cloning with model-based reinforcement learning. The method consists of two stages: (1) anchor-guided behavior cloning for policy initialization and (2) TD3+BC refinement within a single-step learned world model. Experiments on nuScenes and 3DGS simulations demonstrate improved planning accuracy and safety, outperforming behavior cloning and prior state-of-the-art approaches.

Avoiding online exploration reduces deployment risks and costs in safety-critical settings like urban driving. Our framework enables scalable policy learning from logged data and supports generalization to diverse scenarios. However, ensuring unbiased training data and maintaining interpretability remain essential for safe real-world deployment.

Future work includes (1) extending the world model to multi-step prediction, (2) exploring online fine-tuning in simulation or reality, and (3) enriching training data with rare but critical events to boost robustness (Lu et al., 2022; Zhou et al., 2022; Shalev-Shwartz et al., 2016). This study underscores the promise of combining BC and offline RL for safe, scalable deployment without requiring online interaction.

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

# A   APPENDIX

We describe the training setup and architectural configurations for the three core components of our system: Anchor-guided Behavior Cloning, the World Model, and Offline Reinforcement Learning. All experiments are conducted using PyTorch on a machine with 8 NVIDIA RTX 4090 GPUs.

**Anchor-guided Behavior Cloning**
Our behavior cloning (BC) policy predicts velocity-normalized anchor points and heading angles using multi-modal inputs, including BEV features, ego history, and control signals. Each input is processed through a dedicated MLP, then fused into a shared representation.

The network architecture includes a 256-dimensional BEV encoder, two 32-dimensional MLP towers for trajectory and control features, and a three-layer fusion MLP that outputs anchor positions and angles in a multi-task setup.

Training uses a combined position and angle loss, with sine-cosine encoding for angular targets to address periodicity. We adopt the Adam optimizer with mixed-precision training and cosine learning rate scheduling. The model is trained for 50 epochs with a batch size of 128, and performance is evaluated using L2 trajectory error across multiple time horizons.

**World Model**
Our model adopts an encoder–transformer–decoder architecture. The input BEV feature, optionally downsampled via $1 \times 1$ convolution if its channel dimension exceeds 256, is first processed by two convolutional encoder blocks with residual connections and attention, reducing the spatial size from $100 \times 100$ to $25 \times 25$.

Flattened features are fused with a relative motion embedding and a learnable positional embedding before being passed into a 3-layer Transformer encoder with 8 heads and GELU (Hendrycks & Gimpel, 2016) activation. The output sequence is reshaped and upsampled via two decoder blocks to recover spatial resolution. The final output is produced by a lightweight convolutional head and optionally combined with the input BEV via residual addition.

We train the model for 100 epochs using AdamW (Loshchilov & Hutter, 2019) (learning rate 3e-4, weight decay 0.01) with gradient accumulation (step size 4) and cosine learning rate scheduling with warm-up. Mixed precision and multi-GPU support are enabled. A combined MSE and Smooth L1 loss is used to balance accuracy and robustness.

**Offline Reinforcement Learning**
We refine the policy initialized by anchor-guided BC using a TD3+BC strategy under a learned one-step world model. The critics take BEV state $s$ and ego-action $a$ as input, then invoke the world model $\mathcal{M}$ to predict the next state $\hat{s}' = \mathcal{M}(s, a)$. The reward $r = \mathcal{R}(\hat{s}')$ is derived from this predicted state, e.g., via measuring goal progress or violation risk. Two critic networks $Q_1$ and $Q_2$ are trained to minimize Bellman error using the minimum of the twin $Q$ targets with target policy smoothing and clipped Gaussian noise.

The actor network is updated every few steps using both a policy gradient (from $Q_1$) and a behavior cloning loss, weighted by $\lambda$. The final objective encourages the policy to maximize predicted returns while staying close to the offline demonstrations. Actions are normalized by estimated velocity before critic evaluation. Training employs mixed-precision (AMP), Huber loss for stability, L2 regularization, gradient clipping (0.5), and soft target updates ($\tau = 0.005$). Learning rates are decayed every 10K steps by 0.5. We use Adam optimizers with initial learning rates of $1\mathrm{e}{-4}$ for critics and $3\mathrm{e}{-4}$ for the policy. Critic and actor modules are fully decoupled, and all target networks are initialized via hard copy at startup.

