# OpenReview forum: "Anchor-Guided Behavior Cloning with Offline Reinforcement Learning for Robust Autonomous Driving"
_ICLR.cc/2026/Conference — ICLR 2026 Conference Withdrawn Submission_

### Official Review · Reviewer_jn4G · 2025-10-15

**Soundness:** 2
**Presentation:** 3
**Contribution:** 1
**Rating:** 2
**Confidence:** 5

**Summary:**

The paper introduce a new E2E autonomous driving method which first learns a anchor-based trajectory prediction policy by BC and then finetuning it using a world model with single-step offline reinforcement learning. The paper achives state-of-the-art performance on the nuScenes and test it in a realistic simulator built with 3D Gaussian Splatting.

**Strengths:**

1. The paper test it in a realistic simulator built with 3D Gaussian Splatting to demonstrate its closed-loop performance.
2. It achives state-of-the-art performance in the nuScenes dataset.
3. The paper is well written and easy to follow.

**Weaknesses:**

1. The anchor-based trajectory prediction method is not novel in E2E autonomous driving. The RL finetuning method also has been explored in RAD and there exists a lot of works using world model for finetuning. There is no comparision with these work.
2. The paper only compare the performance on the simple nuScenes dataset. The nuScenes dataset has a lot of straight driving scenarios which maybe the reason why the constant velocity anchor works. Considering evaluating on more benchmarks such as Navsim and Bench2Drive, CARLA v2 with more comprehensive driving-related metrics.
3. The detail of the realistic simulator built with 3D Gaussian Splatting is missing, which makes the comparision not convincing.
4. The detail of reward design is missing.
5. The performance of ABC-RL on nuScenes dataset is missing.

**Questions:**

1. Why does not use the world model for online RL finetuning instead of offline RL?
2. Address the weakness.

---

### Official Review · Reviewer_pgEh · 2025-10-25

**Soundness:** 3
**Presentation:** 3
**Contribution:** 2
**Rating:** 4
**Confidence:** 4

**Summary:**

This paper presents ABC-RL, a hybrid framework combining Anchor-guided Behavior Cloning (ABC) and offline Reinforcement Learning (RL) under a learned single-step world model to improve the robustness of autonomous-driving policies trained from logged data.
The main contributions are:

1. Anchor-guided Behavior Cloning (ABC): Introduces dynamics-aware intermediate trajectory targets (anchor points) that normalize trajectory prediction by ego-velocity, improving generalization across speeds and driving styles.

2. World Model Integration: Learns a single-step BEV-based world model that predicts the next state and reward from offline data, enabling model-based policy refinement without online interaction.

3. Offline RL Refinement: Adopts TD3+BC with conservative regularization for policy fine-tuning under the learned dynamics, balancing imitation and reward optimization.

Overall, the work targets the persistent gap between open-loop imitation and closed-loop deployment, proposing a framework for robust, data-driven driving policy learning.

**Strengths:**

1. The paper propose anchor-guided BC. Using velocity-normalized anchor trajectories provides a mechanism to decouple motion dynamics from scene semantics, addressing one of the key limitations of behavior cloning.

2. Integrating offline RL via a learned world model (TD3+BC on synthetic rollouts) is conceptually coherent, representing a fresh perspective on combining world-model-based policy learning and trajectory-level imitation. The methodology is technically sound and builds upon established frameworks (e.g., Dreamer, TD3+BC, VAD) in a consistent way.

3. Experiments are comprehensive, covering open-loop, world-model validation, and closed-loop simulation. The reported gains — especially the 57% reduction in collision rate — are substantial and clearly linked to the proposed components.

4. The paper is clearly written.

**Weaknesses:**

1. The whole page 6 and part in page 7 are not the contribution of this work. It's just the description for TD3+BC. I don't know why waste space discussing this.

2. The “single-step world model” only supports short-horizon rollouts, which constrains the potential benefits of model-based RL. Extending to multi-step latent dynamics (e.g., DreamerV3-style) could strengthen the results.

3. The paper briefly mentions that rewards are inferred from predicted BEV features (goal progress, collision risk) but does not specify their formulation or calibration. Clarifying how these are derived and scaled is crucial to understanding policy improvement stability. We only have "The reward r = R(ˆs′) is derived from this predicted state, e.g., via measuring goal progress or violation risk." in the paper.

4. The constant velocity baseline is not provided. We can't access how well the model does in predicting offset, or the performance gain is comping from the heuristic that the anchor is computed from current velocity.

5. The detailed analysis on the offline RL is missing.

6. Why don't you put the Anchor-BC-RL result in Table 1? Does RL harm open-loop performance? Why? Analysis is expected.

7. People have tried doing GRPO with collision reward to improve planner's performance. I don't see the comparison with simple online RL baseline.


[1] Poutine: Vision-Language-Trajectory Pre-Training and Reinforcement Learning Post-Training Enable Robust End-to-End Autonomous Driving

**Questions:**

1. How to make sure the world model is properly learned and the predicted reward is correct?

2. What will happen if the velocity is zero? Then how you initialize anchors?

3. Why you define the offset against anchor by the dx and dy in the ego car's current frame? Have you tried using the dx and dy rotated by the direction of the velocity?

4. How sensitive is the method to anchor horizon length, the number of anchors, and normalization choice?

5. Can you post the result of constant velocity baseline? (basically just set the offset dx and dy to zero and don't let model predict the offset). This is important baseline.

6. What is Retention in Table 2? It's the performance relative to the VAD-Tiny. This is weird that just put some percentages to a table that suppose to be mAP/NDS. Why don't use VAD-Tiny as the world model as it performs better?

7. The offline RL part is very weak in my understanding. Why don't you put the Anchor-BC-RL result in Table 1? Does RL harm open-loop performance? Why? Analysis is expected. The reward function is not provided. People have tried doing GRPO with collision reward to improve planner's performance. Is there any justification for doing offline RL (given that you are using a single-step WM) instead of online RL (GRPO)?

---

### Official Review · Reviewer_UzmP · 2025-10-30

**Soundness:** 2
**Presentation:** 3
**Contribution:** 2
**Rating:** 4
**Confidence:** 4

**Summary:**

The paper proposes ABC-RL, a framework for end-to-end trajectory planning in autonomous driving. ABC-RL integrates two key mechanisms: Anchor-Guided Behavior Cloning (ABC) and world model-based offline reinforcement learning. The ABC mechanism uses dynamics-aware anchor points as intermediate trajectory targets, which helps normalize trajectories across different speeds and driving styles. This reduces imitation errors when compared to traditional techniques like waypoint prediction. The second component, the world model, predicts state transitions and enables offline reinforcement learning, which allows for policy improvement without requiring real-time environmental interaction. The paper demonstrates that ABC-RL significantly outperforms previous methods on the nuScenes benchmark, achieving substantial reductions in trajectory error and closed-loop collision rates.

**Strengths:**

1. The introduction of dynamics-aware anchor points is a valuable innovation. This approach allows for improved trajectory prediction accuracy, especially in handling varying speeds and driving styles, which is crucial for reliable autonomous driving behavior.

2. The world model-based offline reinforcement learning process effectively mitigates the challenges of closed-loop driving by enabling the policy to improve without requiring expensive online interactions. This approach significantly reduces collision rates in closed-loop deployment, which is vital for safety in autonomous driving systems.

**Weaknesses:**

1. The authors present a key visualization of their method in Figure 5, but it appears to be overly clustered. The clarity of this figure could be improved, perhaps by breaking it into smaller, more focused visuals or providing clearer annotations to highlight the critical aspects of the comparison.

2. While the evaluation on the nuScenes dataset is thorough, the authors should consider expanding their analysis to include open-loop benchmarks like NAVSIM [1]. The nuScenes dataset may not fully capture the diversity of driving scenarios [2]. Imitation errors and collision rates are also not sufficient to evaluate the planning performance. On the other hand, NAVSIM provides a more balanced data distribution with a comprehensive set of open-loop metrics that could better highlight the method's generalization ability.

3. The paper focuses primarily on collision rates in closed-loop evaluations. While this is an important safety metric, benchmarks like Bench2Drive [3] also assess other driving behaviors, such as route completion and road adherence. It would be more beneficial for the authors to include these metrics in closed-loop evaluations to provide a more holistic view of the system’s driving performance.

[1] Dauner D, Hallgarten M, Li T, et al. Navsim: Data-driven non-reactive autonomous vehicle simulation and benchmarking[J]. Advances in Neural Information Processing Systems, 2024, 37: 28706-28719.

[2] Li Z, Yu Z, Lan S, et al. Is ego status all you need for open-loop end-to-end autonomous driving?[C]//Proceedings of the IEEE/CVF Conference on Computer Vision and Pattern Recognition. 2024: 14864-14873.

[3] Jia X, Yang Z, Li Q, et al. Bench2drive: Towards multi-ability benchmarking of closed-loop end-to-end autonomous driving[J]. Advances in Neural Information Processing Systems, 2024, 37: 819-844.

**Questions:**

Please refer to the weaknesses section.

---

### Official Review · Reviewer_Qqhy · 2025-10-31

**Soundness:** 3
**Presentation:** 2
**Contribution:** 2
**Rating:** 2
**Confidence:** 5

**Summary:**

This research tackles the challenge of learning robust driving policies from logged data by bridging the gap between training and deployment environments. The proposed framework, ABC-RL, combines anchor-guided behavior cloning with offline reinforcement learning using a learned world model to create more accurate trajectory targets and enable efficient policy refinement. Through extensive evaluation, ABC-RL significantly outperforms existing methods, demonstrating improved planning stability, safety, and reduced errors in both open-loop and closed-loop scenarios.

**Strengths:**

* Usage of Offline RL
  * No need in online data collection
* Usage of the World Model (to help with Offline RL)

**Weaknesses:**

* What simulator is used? 3DGS-based could be any. Need scenes used inside, the number of simulations, etc
  * Moreover, the only metric - collisions per episode - is definitely not enough to assess the method
* The comparison with only VAD / AD_MLP is very questionable
* Unclear how 2D rewards (positive goal-directed and negative unsafe) are defined, and how they are labaled
* No information about $\gamma$ and $\lambda$
* No hypothesis on why Anchor-BC is quite bad in Table 1 for collision rate

**Questions:**

* lines 240-241: should it be "Given $N$ historical actions" instead of trajectories?

---

### Note · Authors · 2025-11-18

I have read and agree with the venue's withdrawal policy on behalf of myself and my co-authors.